# Subcellular Proteomics to Understand Promotive Effect of Plant-Derived Smoke Solution on Soybean Root

**DOI:** 10.3390/proteomes9040039

**Published:** 2021-10-02

**Authors:** Yusuke Murashita, Takumi Nishiuchi, Shafiq Ur Rehman, Setsuko Komatsu

**Affiliations:** 1Faculty of Environment and Information Sciences, Fukui University of Technology, Fukui 910-8505, Japan; yusuke.murashita@gmail.com; 2Research Center for Experimental Modeling of Human Disease, Kanazawa University, Kanazawa 920-8640, Japan; tnish9@staff.kanazawa-u.ac.jp; 3Department of Biology, University of Haripur, Haripur 22620, Pakistan; drshafiq@yahoo.com

**Keywords:** proteomics, soybean, membrane, nucleus, plant-derived smoke solution

## Abstract

Plant-derived smoke solution enhances soybean root growth; however, its mechanism is not clearly understood. Subcellular proteomics techniques were used for underlying roles of plant-derived smoke solution on soybean root growth. The fractions of membrane and nucleus were purified and evaluated for purity. ATPase and histone were enriched in the fractions of membrane and nucleus, respectively. Principal component analysis of proteomic results indicated that the plant-derived smoke solution affected the proteins in the membrane and nucleus. The proteins in the membrane and nucleus mainly increased and decreased, respectively, by the treatment of plant-derived smoke solution compared with control. In the proteins in the plasma membrane, ATPase increased, which was confirmed by immunoblot analysis, and ATP contents increased through the treatment of plant-derived smoke solution. Additionally, although the nuclear proteins mainly decreased, the expression of RNA polymerase II was up-regulated through the treatment of plant-derived smoke solution. These results indicate that plant-derived smoke solution enhanced soybean root growth through the transcriptional promotion with RNA polymerase II expression and the energy production with ATPase accumulation.

## 1. Introduction

Plant-derived smoke is material for promoting plant growth/development and affects plant species from various habitats [1]. Butanolides, including karrikins and cyanohydrin, are the active compounds in plant-derived smoke [1]. Plant-derived smoke positively affected post-germination growth of rice [2,3,4], wheat [5], maize [6], soybean [7,8,9], and chickpea [10]. These findings clarified that stimulatory effects of plant-derived smoke continue in plant growth and enhance the biomass of the crop. Studies on the post-germination of crops treated with plant-derived smoke elucidate that its treatment affects not only the seed-germination stage but also plant growth and development stages [11]. Although the functional mechanisms of plant-derived smoke in the seed-germination stage were clarified with the discovery of karrikin [12], it is not elucidated for plant growth stages.

Research at the protein level of response to plant-derived smoke solution was reported using the proteomic technique, which elucidated proteasome-independent karrikin signaling pathways and refined the emerging model of karrikin action in *Arabidopsis* [13]. In *Arabidopsis*, by analyzing the protein complexes to MAX2 (more axillary growth 2), D14 (dwarf 14), and KAI2 (karrikin insensitive 2), PAPP5 (phytochrome-associated protein phosphatase 5) as a new MAX2 interactor was revealed and provided another link with the light pathway [14]. Proteomic analysis was performed for the identification of mechanisms regulated by plant-derived smoke solution. Proteomic analysis indicated that the proteins related to glycolysis, sucrose synthesis, and signaling were regulated in chickpea [10] and maize [5] by the application of plant-derived smoke solution. However, functional mechanisms at the protein level in crops applied with plant-derived smoke solution are still to be understood.

Plant-derived smoke solution enhanced the growth of soybean seedlings through the ornithine-synthesis and ubiquitin-proteasome pathway [8]. A positive trend of soybean seedlings was identified in flooding-stress tolerance, which involved an increased ATP content and activation of ascorbate/glutathione cycle in response to plant-derived smoke solution [9]. The supply of plant-derived smoke solution during flooding stress led to the change of proteins related to the cell wall and the flooding recovery of soybean after the removal of water [7]. In the present study, to clarify the role of plant-derived smoke solution on soybean root growth, subcellular proteomic analyses were performed. Furthermore, results from proteomic analyses were confirmed with biochemical analyses such as immunoblot analysis and gene-expression analysis.

## 2. Materials and Methods

### 2.1. Plant Material and Treatment

Plant-derived smoke solution was prepared as a previous report [15]. Seeds of soybean (*Glycine max* L. cultivar Enrei) were sterilized with 2% sodium hypochlorite solution, rinsed in water, and sown in silica sand in a seedling case. A total of 200 seeds were sown in each case. Soybeans were grown at 25 °C and 70% humidity under white fluorescent light (160 µmol m^−2^ s^−1^, 16 h light period/day). Two-day-old soybeans were treated with 2000 ppm plant-derived smoke solution for 2 days. As a control, 4-day-old untreated soybeans were used. Root tip was collected as a sample for all experiments. Three independent experiments were performed as biological replications for all experiments, meaning that the seeds were sown on different days.

### 2.2. Isolation of Membrane Fractions

All purification procedures were carried out on ice. Membranes were isolated according to the manufacturer’s instructions of Mem-PER™ Plus Membrane Protein Extraction Kit (Thermo Fisher Scientific, San Jose, CA, USA) with some modifications (Appendix A).

### 2.3. Isolation of Nuclear Fractions

All purification procedures were carried out on ice. Nuclei were isolated according to the manufacturer’s instructions of the Plant Nuclei Isolation/Extraction Kit (Sigma-Aldrich, St. Louis, MO, USA) with some modifications (Appendix A).

### 2.4. Protein Concentration Measurement

The method of Bradford [16] was used to determine the protein concentration with bovine serum albumin as the standard.

### 2.5. Immunoblot Analysis

SDS-sample buffer consisting of 60 mM Tris-HCl (pH 6.8), 2% SDS, 10% glycerol, and 5% dithiothreitol was added to protein extracts [17]. Immunoblot analysis was performed as a previous report [18]. As primary antibodies, anti-ATPase (Agrisera, Vannas, Sweden), anti-histone H3 (Abcam, Tokyo, Japan), anti-calnexin [19], and anti-ascorbate peroxidases (APX) [18] antibodies were used (Appendix A).

### 2.6. Protein Enrichment, Reduction, Alkylation, and Digestion

Quantified proteins (50 µg) were adjusted to a final volume of 100 µL. Protein preparation for proteomic analysis was performed as a previous report [20] (Appendix A).

### 2.7. Protein Identification Using Nano-Liquid Chromatography Mass Spectrometry

The liquid chromatography (LC) conditions as well as the mass spectrometry (MS) acquisition conditions were described in the previous study [7] with some modifications (Appendix A).

### 2.8. MS Data Analysis

The MS/MS searches were carried out using SEQUEST HT search algorithms against the UniprotKB *Glycine*
*max* (SwissProt TreEMBL, TaxID = 3847, version 25 October 2017) using Proteome Discoverer (PD) 2.2 (version 2.2.0.388; Thermo Fisher Scientific). MS data analysis was described in the previous study [7] with some modifications (Appendix A).

### 2.9. Differential Analysis of Proteins Using MS Data

Label-free quantification was also performed with PD 2.2 using precursor ions quantifiler nodes. For differential analysis of the relative abundance of peptides and proteins between samples, the free software PERSEUS (version 1.6.14.0, Max Planck Institute of Biochemistry, Martinsried, Germany) [21] was used. Differential analysis of proteins using MS data as described in the previous study [7] with some modifications (Appendix A).

### 2.10. Measurement of ATP Contents

ATP assay was performed according to the manufacturer’s instructions of ATP Colorimetric/Fluorometric Assay Kit (Biovision, Milpitas, CA, USA) with some modifications (Appendix A).

### 2.11. Reverse Transcriptase-Polymerase Chain Reaction Analysis

A portion (0.5 g) of samples were quickly frozen in liquid nitrogen and ground to powder with a mortar and pestle. Total RNAs were isolated according to the previous procedure [22]. First-strand cDNA was synthesized from total RNA (5 µg) using Superscript II reverse transcriptase (RT) (Stratagene, Cedar Creek, TX, USA). Gene-specific primers for 18S rRNA (X02623) (F 5′-TGATTAACAGGGACAGTCGG-3′; R 5′-ACGGTATCTGATCGTCTTCG-3′) and RNA polymerase II (GLYMA_10G238600) (F 5′-GGATATCGACATGGGGTACG-3′; R 5′-TCAACCATGACAGGTGCATT-3′) were used to amplify 200 and 300 bp regions, respectively. The polymerase chain reaction (PCR) cycling was as follows: 2 min at 95 °C (one cycle); 30 s at 95 °C, 30 s at 60 °C; 1 min at 68 °C (25 cycles); 7 min at 68 °C (one cycle).

### 2.12. Statistical Analysis

The statistical significance of data was analyzed by the Student’s *t*-test. A *p*-value of less than 0.05 was considered statistically significant.

## 3. Results

### 3.1. Purification of Membrane and Nucleus Fractions in Soybean Root Tip

To investigate the subcellular function in soybean root tips treated with plant-derived smoke, membrane and nuclear fractions were purified (Figure 1). Root tips of 4-day-soybeans were collected and purified for each fraction. Proteins extracted from each fraction were evaluated using immunoblot analysis.

In the immunoblot analysis, anti-ATPase antibody and anti-calnexin antibody, which were used as marker proteins of membrane and endoplasmic reticulum, respectively. anti-ATPase antibody and anti-calnexin antibody were cross-reacted with 90 kDa protein and 63 kDa protein, respectively (Figure 2). The relative intensity of the corresponding signal band significantly increased in the membrane-protein fraction compared to the cytosolic fraction (Figure 2). To assess protein contamination from other organelles, protein abundance of other subcellular markers in the cytosolic protein and purified membrane-protein fractions were examined (Figure 2). Mitochondrial APX and cytosolic APX were used as marker proteins of mitochondria and cytosol, respectively (Figure 2). The cytosolic fraction had a significantly higher accumulation of cytosolic APX compared with the enriched membrane fractions (Figure 2).

In the immunoblot analysis, an anti-histone H3 antibody as a marker for nuclear protein cross-reacted with a 16 kDa protein (Figure 3). The relative intensity of the corresponding signal band significantly increased in the nuclear-protein fraction compared to the cytosolic fraction (Figure 3). To assess protein contamination from other organelles, protein abundance of other subcellular markers in the cytosolic protein and purified nuclear-protein fractions were examined (Figure 3). Mitochondrial APX and cytosolic APX were used as marker proteins of mitochondria and cytosol, respectively (Figure 3). The cytosolic protein fraction had a significantly higher accumulation of cytosolic APX compared with the enriched nuclear fractions (Figure 3). Based on the immunoblot analysis of subcellular-specific proteins, the fractions containing membrane and nuclei were confirmed to be highly enriched for membrane and nuclear proteins and were not so much contaminated with proteins from the cytosol.

### 3.2. Membrane Proteomics of Soybean Treated with Plant-Derived Smoke

To identify membrane proteins affected by plant-derived smoke solution in soybean, a gel-free/label-free proteomic technique was used. Proteins were extracted from root tip of 4-day-old soybeans treated without or with plant-derived smoke solution for 2 days. The digested membrane proteins were analyzed using nanoLC−MS/MS (Thermo Fisher Scientific, Waltham, MA, USA) (Figure 1). The identified proteins of soybean treated with plant-derived smoke solution were compared with those of untreated soybean using SIEVE software (Thermo Fisher Scientific). The proteomic data were estimated using principal component analysis (PCA), which indicated the difference between control and treatment (Figure 4A). To assess the quality of the proteomic data, quality control analysis was performed. The median intra-group coefficient of variation distribution for each sample was less than 10% (Appendix A).

Proteins with at least two matched peptides and a *p*-value less than 0.05 were identified (Appendix A). A total of 220 and 48 (Appendix A) proteins significantly increased and decreased, respectively, by treatment of plant-derived smoke solution compared to control. To better understand the function of the identified proteins, the differentially accumulated proteins were subjected to a gene ontology (GO) database (Figure 4B). Protein abundance in the peroxisome, mitochondria, chloroplast, endoplasmic reticulum, and plasma membranes mainly increased by treatment of plant-derived smoke solution compared with control (Figure 4B). In the proteins of the membrane, three catalase (B0M1A4, O48560, and P29756) and peroxidase (A0A0R0GXD9) significantly decreased (Appendix A).

### 3.3. Nuclear Proteomics of Soybean Treated with Plant-Derived Smoke

To identify nuclear proteins affected by plant-derived smoke solution in soybean, a gel-free/label-free proteomic technique was used. Proteins were extracted from the root tip of 4-day-old soybeans treated without or with plant-derived smoke solution for 2 days. After digestion, nuclear proteins were analyzed using nanoLC-MS/MS (Figure 1). The identified proteins of soybean treated with plant-derived smoke solution were compared with those of untreated soybean using SIEVE software. The proteomic data were estimated using PCA, which indicated the difference between control and treatment (Figure 5A). To assess the quality of the proteomic data, a quality control analysis was performed. The median intra-group coefficient of variation distribution for each sample was less than 10% (Appendix A).

Proteins with at least two matched peptides and a *p*-value less than 0.05 were identified (Appendix A). A total of 29 and 144 (Appendix A) proteins significantly increased and decreased, respectively, by treatment of plant-derived smoke solution compared to control. To better understand the function of the identified proteins, the differentially accumulated proteins were subjected to the GO database (Figure 5B). In the nuclear proteins, Root hair initiation protein (I1L3V3) and RNA polymerases II (I1LDU4) significantly increase by plant-derived smoke solution compared to control (Table 1). On the other hand, four late embryogenesis-abundant (LEA) proteins (Q9XET0, I1L957, I1M3M9, and I1LE41), importin (I1MDN4), histone acetyltransferase (C6SWV3) significantly decreased by plant-derived smoke solution compared to control (Table 1).

### 3.4. ATPase Abundance and ATP Contents in Soybean Treated with Plant-Derived Smoke

Based on proteomic results, some of the proteins were confirmed on the abundance level using biochemical techniques. To identify the function of the plasma membrane, the abundance of ATPase was analyzed using the immunoblot technique. The abundance of ATPase increased by treatment of plant-derived smoke solution (Figure 6A and Appendix A). Additionally, ATP contents were analyzed, and it increased by treatment of plant-derived smoke solution (Figure 6B). The proteins in the plasma membrane mainly increased (Figure 4B), and ATPase increased, which was confirmed by immune-blot analysis, as well as ATP contents increased by the treatment of plant-derived smoke solution (Figure 6).

### 3.5. Expression of RNA Polymerase II in Soybean Treated with Plant-Derived Smoke

Based on proteomic results, the expression level was confirmed with RT-PCR analysis. RT-PCR analysis of RNA polymerase II in soybean treated with plant-derived smoke solution was performed (Figure 7). RNA polymerase II-specific oligonucleotides were used to amplify transcripts of total RNA isolated from soybean root, and 18S rRNA was used as an internal control. The expression level of 18S rRNA did not change between without and with plant-derived smoke solution. However, RNA polymerase II was significantly up-regulated by the treatment of plant-derived smoke solution (Figure 7). Although the nuclear proteins mainly decreased (Figure 5B), the expression of RNA polymerase II was up-regulated by the treatment of plant-derived smoke solution (Figure 7).

## 4. Discussion

### 4.1. The Effect of Plant-Derived Smoke Solution on Membrane in Soybean Root

To identify for the first cell-regulatory event controlling proteins in soybean root tip through plant-derived smoke solution, membrane-proteomic analysis, which can detect low abundant proteins, was performed. In this study, three catalase (B0M1A4, O48560, and P29756) and peroxidase (A0A0R0GXD9) decreased (Appendix A) by plant-derived smoke solution. The enzymatic antioxidative system is an essential mechanism for fine-tuning the regulation of the level of reactive oxygen species (ROS) [23,24]. ROS produced in plants is generally a superoxide anion, which is immediately converted to a less reactive form of ROS and hydrogen peroxide by the action of superoxide dismutase. The superoxide dismutase functions as the first line of defense against ROS [25]. Ascorbate peroxidase and catalase subsequently detoxified the resulting hydrogen peroxide [26]. Excess soil moisture conditions definitively increased oxidative damage through the decreased activity of antioxidative enzymes and the consequent ROS accumulation at both the seedling and vegetative stages. Additionally, increased antioxidative enzyme activity enhanced scavenging excess ROS, reducing oxidative damage, and improving the tolerance to excess moisture [27,28]. This result with previous findings suggests that plant-derived smoke solution might decrease ROS scavenging enzyme through reduction in stress such as oxidative stress.

Proteins related to the mitochondrial electron transport chain decreased under flooding stress. In contrast, these same proteins increased under the same conditions through the addition of plant-derived smoke solution [9]. In this study, to understand the energy changes in soybean root growth treated with plant-derived smoke, plasma membrane ATPase abundance, and ATP content were measured. Even when there was no flooding stress, ATPase abundance and ATP content increased with the treatment of plant-derived smoke solution (Figure 7). ATP contents were higher in the actively growing cells of cryotolerant algae [29]. The growth and development of living organisms were linked to proper photosynthetic function, which is driven by the use of ATP [30]. These results with previous reports suggest that plant-derived smoke solution might enhance soybean growth through the increase in ATP content by activation of ATPase in the root.

### 4.2. The Effect of Plant-Derived Smoke Solution on Nucleus in Soybean Root

In order to identify the upstream-regulatory events that controlled proteins in soybean root tips through plant-derived smoke solution, nuclear proteomic analysis was performed. Transcription of protein-encoding genes starts with forming a pre-initiation complex comprised of RNA polymerase II and several general transcription factors [31]. RNA polymerase II modifications play a critical biological function in fundamental cellular processes. The C-terminal domain of the large subunit of RNA polymerase II contains highly conserved heptad repeats, which are subjected to several post-translational modifications in the transcription cycle [32]. Histone acetyltransferase GCN5 is involved in heat-stress responses, and it targets the heat-stress response genes, heat-stress transcription factors 3, and UV-hypersensitive 6. Histone acetyltransferase GCN5 induced their expression, increasing RNA polymerase II engagement and H3K9/14 acetylation levels [33]. In this study, RNA polymerase was up-regulated (Figure 7); however, histone acetyltransferase decreased in the nucleus through treatment of plant-derived smoke (Table 1). These results with previous findings suggest that plant-derived smoke might be up-regulated by plant-derived smoke, but protein modification is not vital for its function.

LEA proteins are localized in all cellular compartments, such as the nucleus [34], chloroplast [35], mitochondria [36], endoplasmic reticulum [37], vacuole [38], cytoplasm [39], and plasma membrane [40], where they are proposed to exert their protective function acting as chaperones to repair improperly folded proteins [41]. LEA proteins accumulated in the nucleus [42] or were translocated from the cytosol to the nucleus upon phosphorylation [43]. LEA proteins during desiccation tolerance reveal their role in the transportation of nuclear-targeted proteins during stress [44], stabilizing membrane structures as antioxidants by binding to metal ions, increasing cellular mechanical strength by the generation of filaments [45], and the protection of cell membranes against desiccation [46]. In this study, 4 LEA proteins (Q9XET0, I1L957, I1M3M9, and I1LE41) decreased in the nucleus. On the other hand, several LEA proteins increased in the membrane (Appendix A). These results with previous reports suggest that LEA proteins might transfer from nucleus to cytosol and/or to membrane from cytosol with a treatment of plant-derived smoke solution.

## 5. Conclusions

Plant-derived smoke solution enhances soybean root growth [8]. To understand the role of plant-derived smoke solution on soybean root growth, subcellular proteomic techniques were used. The main findings are as follows: (i) the membrane and nucleus of soybean root tips were enriched in each fraction; (ii) PCA indicated that plant-derived smoke solution affected proteins in the membrane and nucleus; (iii) in the plasma membrane, ATPase and ATP contents increased through plant-derived smoke solution; (iv) although the nuclear proteins mainly decreased, RNA polymerase II was expressed through plant-derived smoke solution. These results suggest that plant-derived smoke solution improves the root growth of soybean through the transcriptional promotion with RNA polymerase II expression and the energy production with ATPase accumulation.

## Figures and Tables

**Figure 1 proteomes-09-00039-f001:**
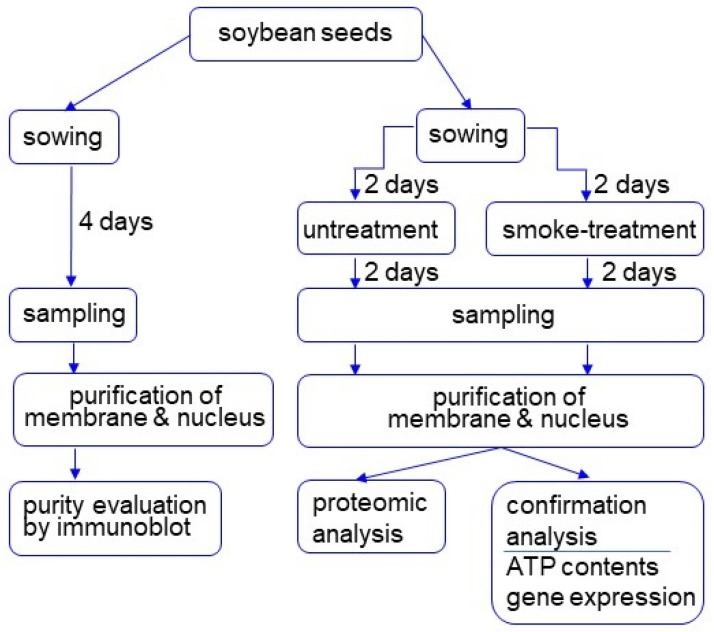
Experimental design for investigation of the subcellular mechanism in soybean treated with plant-derived smoke solution. Two-day-old soybeans were treated with plant-derived smoke solution for 2 days, and membrane and nuclear fractions were purified. After purity evaluation, proteins were analyzed using proteomic techniques. Furthermore, proteomic results were confirmed with biochemical experiments. All experiments were performed with 3 independent biological replicates.

**Figure 2 proteomes-09-00039-f002:**
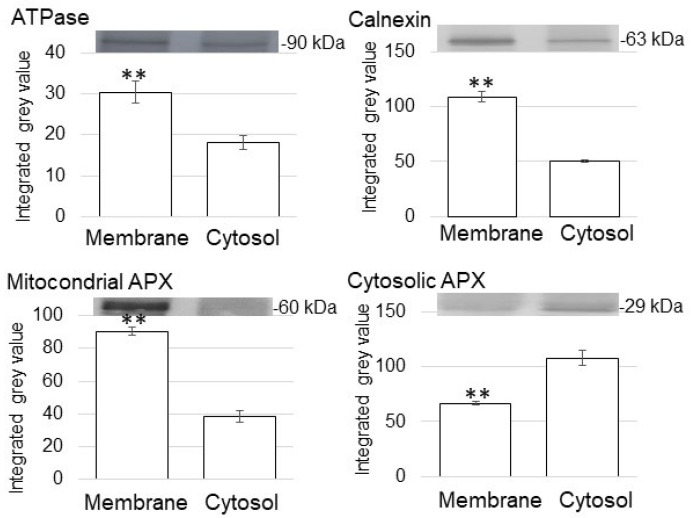
Purity of enriched membrane fractions obtained from soybean root tip. Proteins were extracted from crude-membrane fraction and analyzed the purity. The obtained membrane-protein fraction and cytosolic protein fraction were analyzed using immunoblot analysis. The abundance of ATPase as a marker protein for membrane, calnexin as a marker protein for endoplasmic reticulum membrane, mitochondrial APX, and cytosolic APX were used. Data are shown as means ± SD from 3 independent biological replicates. Asterisks indicate significant changes in the relative intensity of signal band in the membrane fraction compared to cytosolic fraction according to Student’s *t*-test (**: *p* < 0.01).

**Figure 3 proteomes-09-00039-f003:**
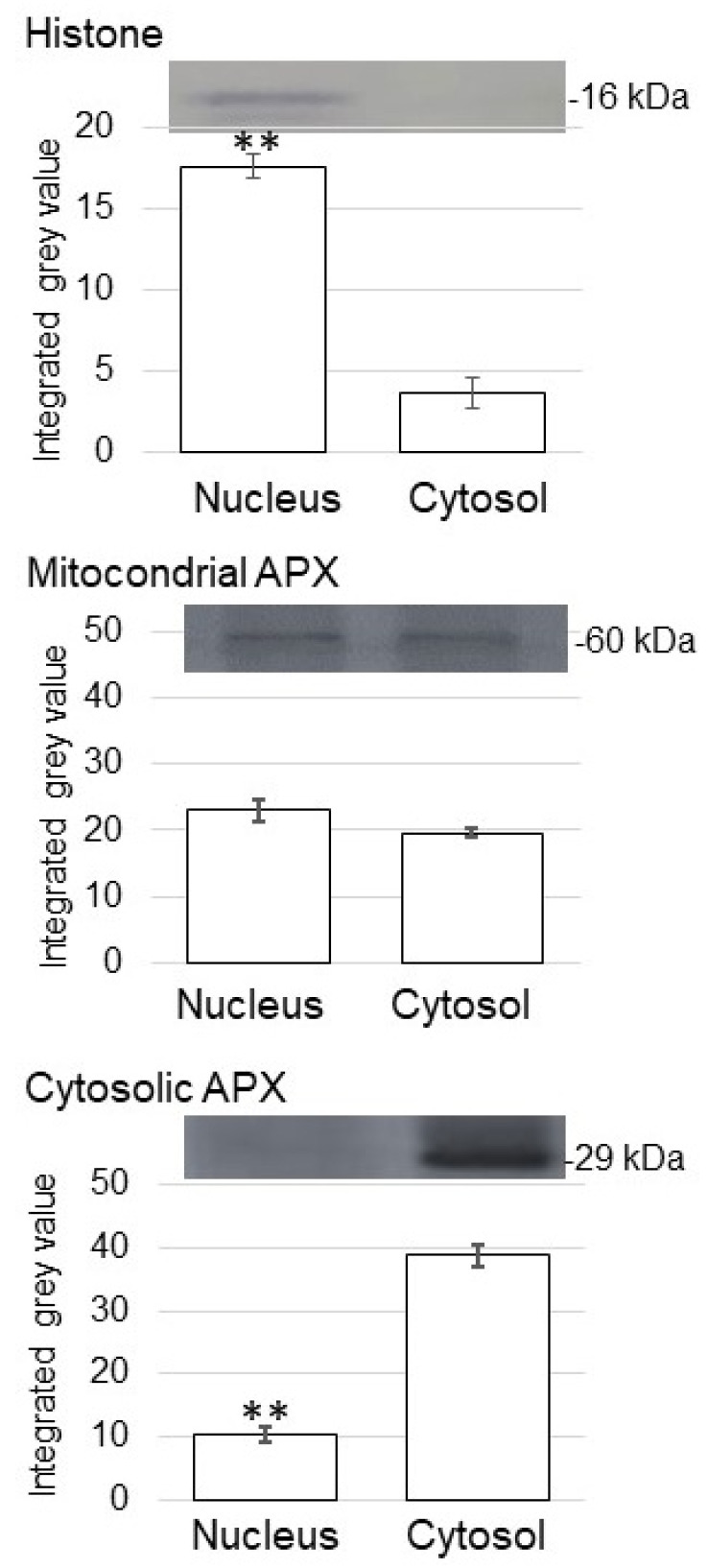
Purity of enriched nuclear fractions obtained from soybean root tip. Proteins were extracted from crude-nuclear fraction and analyzed the purity. The obtained nuclear-protein fraction and cytosolic protein fraction were analyzed using immunoblot analysis. The abundance of histone as a marker protein for nucleus, mitochondrial APX, and cytosolic APX were used. Data are shown as means ± SD from 3 independent biological replicates. Asterisks indicate significant changes in the relative intensity of signal band in the nuclear fraction compared to cytosolic fraction according to Student’s *t*-test (**: *p* < 0.01).

**Figure 4 proteomes-09-00039-f004:**
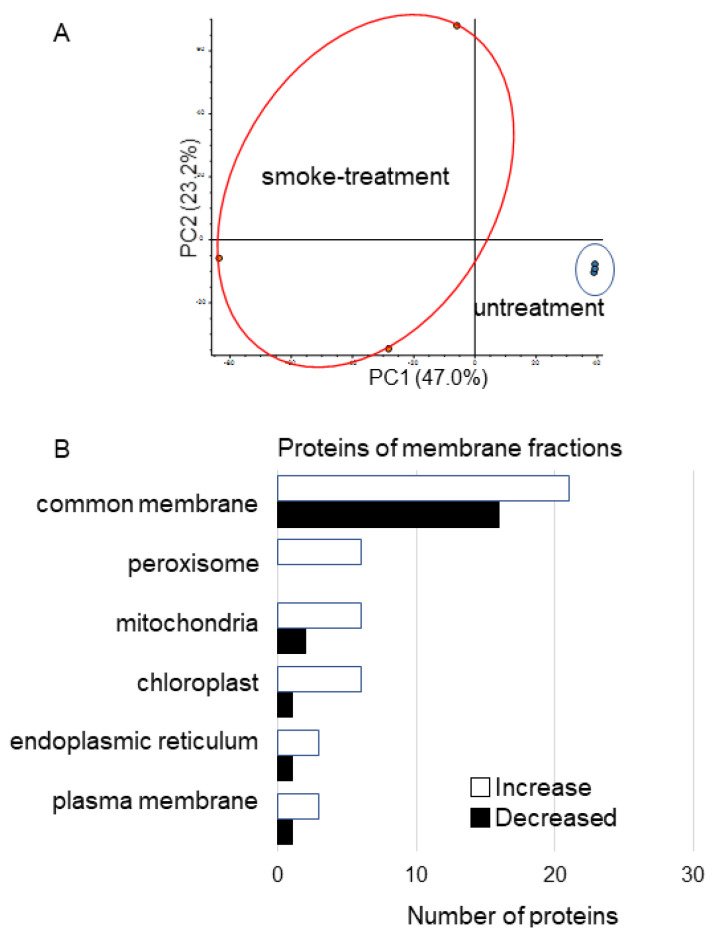
Overview of total proteomic data and functional categories of membrane proteins in soybean treated with plant-derived smoke solution. (**A**) Overview of total proteomic data from 6 samples of membrane fractions of soybean based on PCA. Proteomic analysis was performed with 3 independent biological replicates for each treatment. PCA was performed with PD 2.2. (**B**) Functional categories of proteins with differential abundance in membrane fraction in soybean. After proteomic analysis, significantly changed proteins (*p* < 0.05) in soybean-membrane fraction were compared between with and without plant-derived smoke solution. Functional categories of changed proteins were determined using GO analysis. White and black columns show increased and decreased proteins, respectively.

**Figure 5 proteomes-09-00039-f005:**
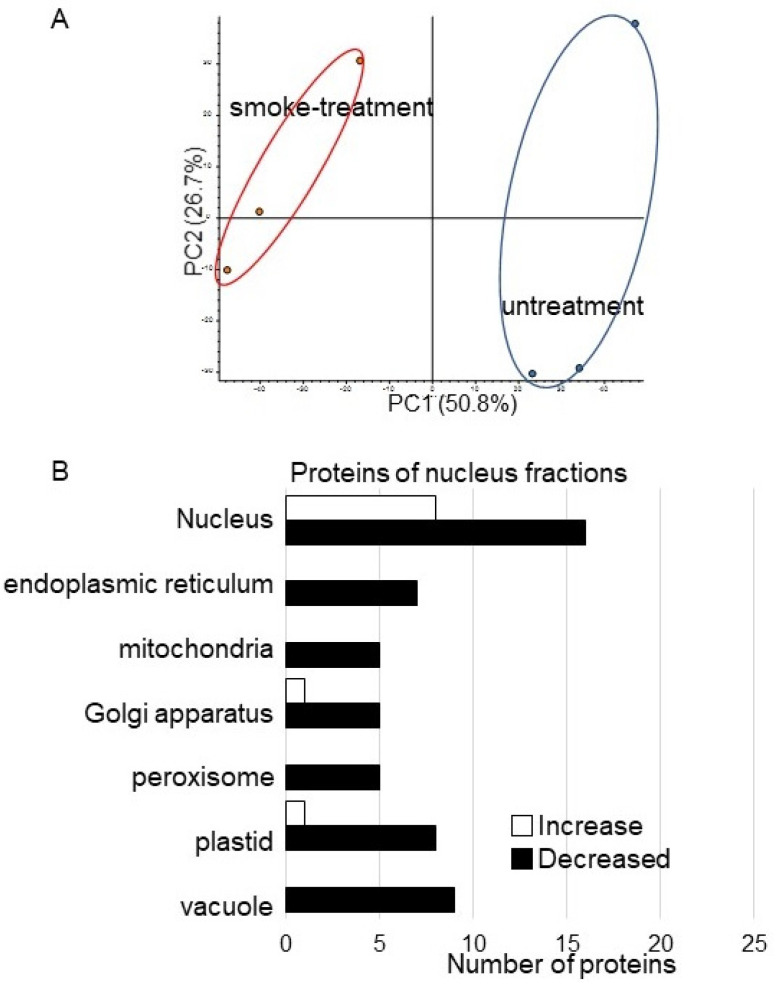
Overview of total proteomic data and functional categories of nuclear proteins in soybean treated with plant-derived smoke solution. (**A**) Overview of total proteomic data from 6 samples of nuclear fractions of soybean based on PCA. Proteomic analysis was performed with 3 independent biological replicates for each treatment. PCA was performed with PD 2.2. (**B**) Functional categories of proteins with differential abundance in a nuclear fraction in soybean. After proteomic analysis, significantly changed proteins (*p* < 0.05) in soybean-nuclear fraction were compared between with and without plant-derived smoke solution. Functional categories of changed proteins were determined using GO analysis. White and black columns show increased and decreased proteins, respectively.

**Figure 6 proteomes-09-00039-f006:**
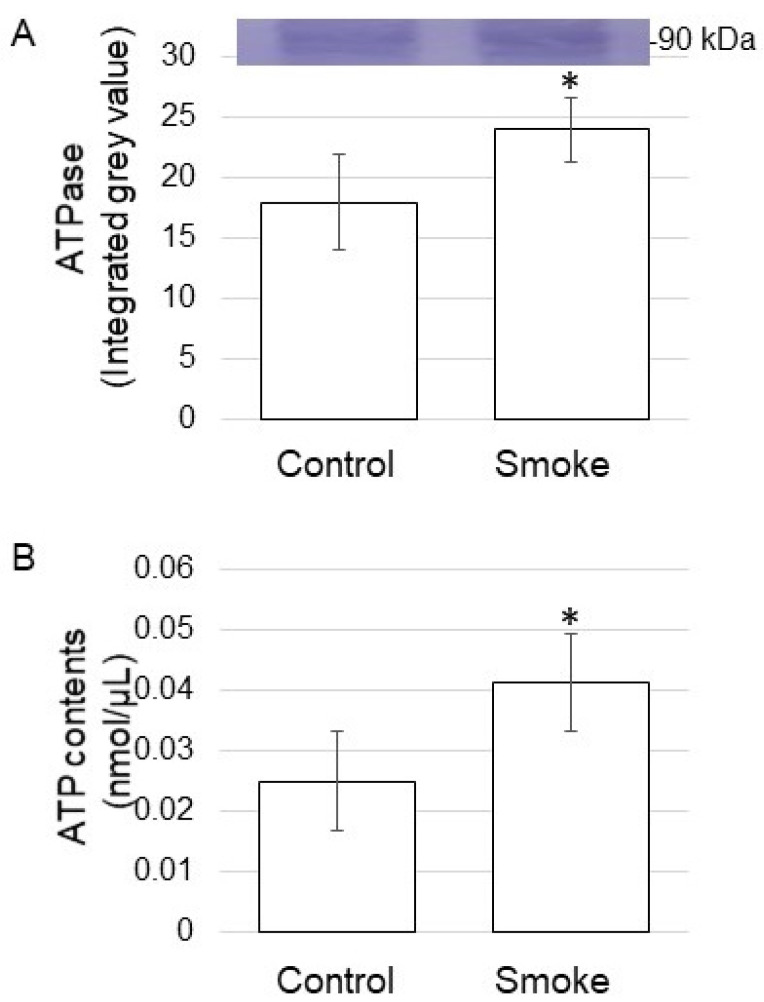
ATPase abundance and ATP contents of soybean treated with plant-derived smoke solution. Two-day-old soybeans were treated without (Control) or with plant-derived smoke solution (Smoke) for 2 days. (**A**) Proteins (10 μg) extracted from the root were separated on SDS-polyacrylamide gel and transferred on to polyvinylidene difluoride membrane. The polyvinylidene difluoride membrane was cross-reacted with an anti-ATPase antibody. (**B**) Metabolites were extracted from the root, and ATP contents were measured for each sample. Data are shown as means ± SD from 3 biological replicates (Appendix A). Student’s *t*-test was used to compare values between control and smoke groups. Asterisk indicates a significant change (* *p* ≤ 0.05).

**Figure 7 proteomes-09-00039-f007:**
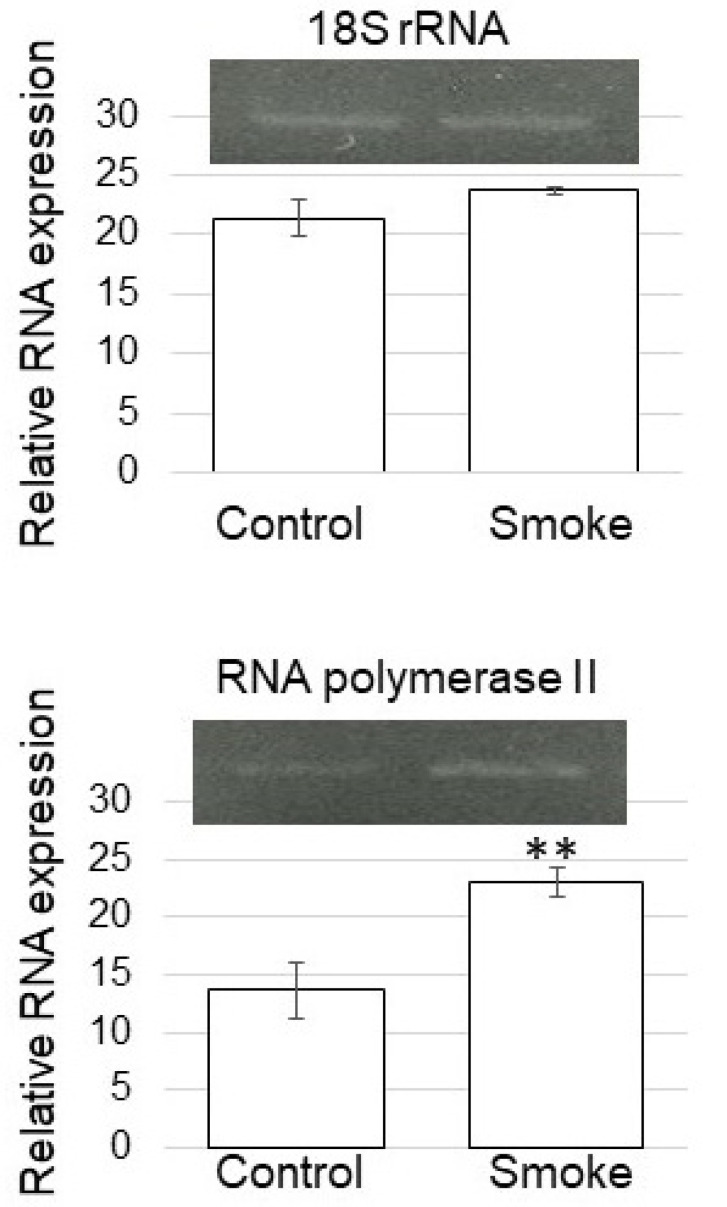
Gene expression of RNA polymerase II of soybean treated with plant-derived smoke solution. RT-PCR analysis of RNA polymerase II in soybean treated without (Control) or with plant-derived smoke solution (Smoke) was performed. RNA polymerase II-specific oligonucleotides were used to amplify transcripts from total RNA isolated from soybean root. 18S rRNA was used as an internal control. Data are shown as means ± SD from 3 biological replicates. Student’s *t*-test was used to compare values between control and smoke groups. Asterisk indicates a significant change (** *p* ≤ 0.01).

**Table 1 proteomes-09-00039-t001:** List of changed nuclear proteins in soybean root tip treated with plant-derived smoke solution compared with control.

No	Accession Number	Description	Matched Peptides	Ratio
Increased				
1	I1L3V3	Root hair initiation protein root hairless	3	100
2	I1LDU4	RNA polymerases II	2	100
3	C6TGY7	Proliferating cell nuclear antigen	3	29.341
4	I1LHP2	Tyrosyl-tRNA synthetase/Nucleotidylyl transferase	2	5.495
5	I1JJS2	WD repeats region domain-containing protein	3	5.458
6	C6TAF1	Nucleolar essential protein-like protein	2	5.391
7	I1JHW9	CBF domain-containing protein	10	5.143
8	C6TGI7	Leucine-rich repeat (LRR) family protein	2	5.688
Decreased				
1	I1LP68	Bet v 1 domain-containing protein	2	0.174
2	K7LQ69	Poly (ADP-ribose) polymerase	4	0.155
3	Q9SWB4	Protein ADP-ribosyltransferase PARP3	3	0.126
4	Q9XET0	Seed maturation protein PM30	3	0.123
5	I1L957	Late embryogenesis abundant protein,	6	0.051
6	I1JFX0	Usp domain-containing protein	7	0.022
7	I1JFX0	Usp domain-containing protein	7	0.022
8	I1M3M9	Late embryogenesis abundant protein	11	0.010
9	I1L849	Late embryogenesis abundant protein D-34	10	0.010
10	I1LE41	Seed maturation protein	7	0.010
11	C6SWV3	Histone acetyltransferase	4	0.010
12	I1MDN4	Importin N-terminal domain-containing protein	2	0.010

## Data Availability

For MS data, RAW data, peak lists, and result files have been deposited in the ProteomeXchange Consortium [47] via the jPOST [48] partner repository under data-set identifiers PXD016775 (Membrane proteins) and PXD016753 (Nucleus proteins).

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
