# Peer review of "Subcellular Proteomics to Understand Promotive Effect of Plant-Derived Smoke Solution on Soybean Root"

_proteomes, 2021, doi:10.3390/proteomes9040039_

Round 1

Reviewer 1 Report

The authors set out to investigate the role of plant-derived smoke solution and its effect on soybean root. To answer these questions proteomics was applied on fractions of soybean roots. 

Minor concern - the english could be written more clearly and can at time be hard to understand. 

Major concerns that needs to be commented:

In general it would be nice to see the data points with the histograms as it is hard to determine the distribution of the measured variables.

4B/5B - it would be nice to have the variance added here. 

Figure 6 -> it would be nice to see the data behind these two distributions; also that this is significant is hard to see from the error bars.

Tabel 1 - it is not clear to this reviewer how these ratios have been calculated and why some have 3 significant numbers while some have none.

It would be good to fix these points   

Author Response

Reviewer 1

The authors set out to investigate the role of plant-derived smoke solution and its effect on soybean root. To answer these questions proteomics was applied on fractions of soybean roots. 

Minor concern - the english could be written more clearly and can at time be hard to understand. 

Answer: As suggested, this revised article has been corrected by native speaker.

Major concerns that needs to be commented:

In general it would be nice to see the data points with the histograms as it is hard to determine the distribution of the measured variables.

Answer: As suggested, histograms have been added as new supplemental figures 1and 3 for membrane proteins and nuclear proteins, respectively.

4B/5B - it would be nice to have the variance added here. 

Answer: As you known, because this is number of proteins from differential analysis of a proteomic data, which is using Proteome Discoverer, there is no variance. Because protein abundance data have the variance, those data have been added as new supplemental figures 2 and 4 for membrane proteins and nuclear proteins, respectively.

Figure 6 -> it would be nice to see the data behind these two distributions; also that this is significant is hard to see from the error bars.

Answer: Thank you very much for your comments. As suggested, immunoblot data of triplicate experiments have been shown as new supplemental figure 5. Furthermore, the statistical significance of data was analyzed by the Student's t-test. A p-value of less than 0.05 was considered as statistically significant. This information has been clarified in “2.12. Statistical Analysis” and figure legend of Figure 6.

Tabel 1 - it is not clear to this reviewer how these ratios have been calculated and why some have 3 significant numbers while some have none.

Answer: Thank you very much for your question. MS data have been calculated using “Thermo Proteome Discoverer-User Guide (Software 2.2) Page: 541-542”.

Calculating Ratios Using the Pairwise Ratio Approach are as follows: Use this method when the Ratio Calculation parameter in the Precursor Ion Quantifier and the Reporter Ions Quantifier nodes are set to Pairwise Ratio Based. For non-nested study designs, the application calculates the peptide group ratios as the geometric median of all combinations of ratios from all the replicates for the selected study factors. The protein ratio is subsequently calculated as the geometric median of the peptide group ratios. For precursor ion quantification, the pairwise ratios for the peptide groups for the different isotopic forms of a given peptide are combined and displayed on only one of the isotopic forms for that peptide. For a nested study design, the replicates are calculated as the geometric median of all the pairwise ratios for the technical replicates for the given biological replicate. The peptide sample group ratios are then calculated as the geometric median of the biological replicate ratios just as is done for the summed abundance method above. The protein ratio is then calculated as the geometric median of the peptide group ratios.

Reviewer 2 Report

In this manuscript, the author did subcellular proteomics to understand the promotive effect of the plant-derived smoke solution on soybean roots. In this study, the fractions of membrane and nucleus were purified and evaluated for purity. As a result, ATPase and histone were enriched in the fractions of membrane and nucleus, respectively. Principal-component analysis of proteomic results indicated that the plant-derived smoke solution affected the proteins in the membrane and nucleus. The proteins in the membrane and nucleus mainly increased and decreased, respectively, by treating plant-derived smoke solution compared with control. In the proteins in the plasma membrane, ATPase increased, confirmed by immunoblot analysis, and ATP contents increased through the treatment of plant-derived smoke solution. Additionally, although the nuclear proteins mainly decreased, the expression of RNA polymerase II was up-regulated through the treatment of plant-derived smoke solution. These results suggest that plant-derived smoke solution improves the root growth of soybean through the transcriptional promotion with RNA polymerase II expression and the energy production with ATPase accumulation.

The manuscript is heavily plagiarized. It should be clean.

Change at

L12 Subcellular proteomic to Subcellular proteomics.

L40 using proteomic technique to using the proteomic technique.

L51 soybean seedling to soybean seedlings.

L228, L247 fraction had significantly higher to fraction had a significantly higher.

L248 On the basis  of to Based on.

L299 matched peptide to matched peptides.

L324 some of proteins to some of the proteins.

L397 play  critical to play a critical 

This manuscript is heavily plagiarized at L22, L28, L30, L33, L41, L45-48, L51-55, L63-L68, L75-L77, L79-82, L81-85, L86, L91-102, L105-118, L120-125, L126-136, L138, L143-144, L116-153, L158-165, L168-175, L176-178, L181-188, L193-194, L202-204, L222, L224, L235-238, L240-243, L253-254, L265, L267-268, L292-294, L213-214, L233-234, L338-340, L357-360, L367-368, L370-376, L380-383, L386-389, L393, L394-401, L403-404, L409, L411-L418.

Author Response

Reviewer 2

In this manuscript, the author did subcellular proteomics to understand the promotive effect of the plant-derived smoke solution on soybean roots. In this study, the fractions of membrane and nucleus were purified and evaluated for purity. As a result, ATPase and histone were enriched in the fractions of membrane and nucleus, respectively. Principal-component analysis of proteomic results indicated that the plant-derived smoke solution affected the proteins in the membrane and nucleus. The proteins in the membrane and nucleus mainly increased and decreased, respectively, by treating plant-derived smoke solution compared with control. In the proteins in the plasma membrane, ATPase increased, confirmed by immunoblot analysis, and ATP contents increased through the treatment of plant-derived smoke solution. Additionally, although the nuclear proteins mainly decreased, the expression of RNA polymerase II was up-regulated through the treatment of plant-derived smoke solution. These results suggest that plant-derived smoke solution improves the root growth of soybean through the transcriptional promotion with RNA polymerase II expression and the energy production with ATPase accumulation.

The manuscript is heavily plagiarized. It should be clean.

Change at

L12 Subcellular proteomic to Subcellular proteomics.

L40 using proteomic technique to using the proteomic technique.

L51 soybean seedling to soybean seedlings.

L228, L247 fraction had significantly higher to fraction had a significantly higher.

L248 On the basis  of to Based on.

L299 matched peptide to matched peptides.

L324 some of proteins to some of the proteins.

L397 play  critical to play a critical 

Answer: Thank you very much for your kind corrections. This article has been corrected by native speaker. They have been corrected with other corrections. Corrected parts have been marked in red color.

This manuscript is heavily plagiarized at L22, L28, L30, L33, L41, L45-48, L51-55, L63-L68, L75-L77, L79-82, L81-85, L86, L91-102, L105-118, L120-125, L126-136, L138, L143-144, L116-153, L158-165, L168-175, L176-178, L181-188, L193-194, L202-204, L222, L224, L235-238, L240-243, L253-254, L265, L267-268, L292-294, L213-214, L233-234, L338-340, L357-360, L367-368, L370-376, L380-383, L386-389, L393, L394-401, L403-404, L409, L411-L418.

Answer: Thank you very much for your point out. “Materials and method” could not be changed; however, they have been deleted with references and with new supplemental table 1. For other parts, they have been rephrased. Corrected parts have been marked in red color.

Round 2

Reviewer 2 Report

I am happy with the author's reply and the manuscript is much refined now. It can be accepted in its current form.